# Bright and durable scintillation from colloidal quantum shells

Burak Guzelturk [1] ✉, Benjamin T. Diroll [2] ✉, James P. Cassidy[3], Dulanjan Harankahage[3], Muchuan Hua [2], Xiao-Min Lin [2], Vasudevan Iyer[4], Richard D. Schaller [2,5], Benjamin J. Lawrie [4,6] & Mikhail Zamkov [3] ✉

Efficient, fast, and robust scintillators for ionizing radiation detection are crucial in various fields, including medical diagnostics, defense, and particle physics. However, traditional scintillator technologies face challenges in simultaneously achieving optimal performance and high-speed operation. Herein we introduce colloidal quantum shell heterostructures as X-ray and electron scintillators, combining efficiency, speed, and durability. Quantum shells exhibit light yields up to 70,000 photons MeV$^{-1}$ at room temperature, enabled by their high multiexciton radiative efficiency thanks to long Auger-Meitner lifetimes (>10 ns). Radioluminescence is fast, with lifetimes of 2.5 ns and sub-100 ps rise times. Additionally, quantum shells do not exhibit afterglow and maintain stable scintillation even under high X-ray doses (>10$^9$ Gy). Furthermore, we showcase quantum shells for X-ray imaging achieving a spatial resolution as high as 28 line pairs per millimeter. Overall, efficient, fast, and durable scintillation make quantum shells appealing in applications ranging from ultrafast radiation detection to high-resolution imaging.

Scintillators convert high-energy radiation or energetic particles into visible light and play a vital role in broad scientific and technological domains. These include medical diagnostics, geophysical and space exploration, X-ray security, high-energy particle research, and many more. Traditional scintillator technologies, often inorganic ceramics, have long served as efficient radioluminescent emitters. However, these inorganic scintillators face inherent challenges, particularly in achieving fast response times on nanosecond timescales, together with afterglow problems arising from millisecond to second timescale phosphorescence. Furthermore, ceramic scintillators, which are synthesized at high temperatures, cannot be easily scaled up to large areas required for security and imaging applications. In contrast, scalable scintillators based on solution-processed materials are impeded by stability concerns and sub-par performances. Consequently, there is an ongoing search to identify and develop new scintillators that combine all necessary attributes of a practical scintillator: high efficiency, rapid response time, enhanced energy and spatial resolution, longevity, and scalability.

In recent years, colloidal semiconductor nanocrystals have emerged as promising candidates for scintillators. These nanocrystals, renowned for their bright light emission, offer several advantages, such as narrow emission line widths leading to vibrant colors tunable with size, shape and composition control[1,2]. Such colloidal nanocrystals are compatible with solution processing and large-area fabrication techniques[3]. By harnessing the efficient down-conversion capabilities of nanocrystals, high-energy radiation or charged particles could be converted into visible light[4,5], opening avenues for scintillator design. Previous studies have shown nanocrystal scintillators for X-rays[6,7], gamma[8,9], neutrons[10], and electrons[11]. These scintillators have used various nanocrystal compositions such as cadmium chalcogenides[12,13],

[1]X-ray Science Division, Argonne National Laboratory, Lemont, IL, USA. [2]Center for Nanoscale Materials, Argonne National Laboratory, Lemont, IL, USA. [3]Department of Physics, Bowling Green State University, Bowling Green, OH, USA. [4]Center for Nanophase Materials Sciences, Oak Ridge National Laboratory, Oak Ridge, TN, USA. [5]Department of Chemistry, Northwestern University, Evanston, IL, USA. [6]Materials Science and Technology Division, Oak Ridge National Laboratory, Oak Ridge, TN, USA. ✉e-mail: burakg@anl.gov; bdiroll@anl.gov; zamkovm@bgsu.edu

transition metal oxides[14,15], and metal halide perovskites[6,16,17]. Furthermore, polymer-nanocrystal composites have been developed for improved film-forming properties[8,9,17].

Despite the significant potential of nanocrystals as scintillators, several challenges have remained to date. Prior literature suggested the role of multiexcitons in radio- and cathodoluminescence of nanocrystals[13,18], evidenced by sub-nanosecond emission decays[19] that are significantly faster than the lifetime of a single exciton state. While sub-nanosecond radioluminescence is promising for time-of-flight positron emission tomography (TOF-PET)[9,20], scintillation efficiency, quantified by the light yield (LY), remains rather low in these materials because multiexcitons recombine nonradiatively via Auger-Meitner processes[21]. For example, the LY of Cd-chalcogenide nanocrystals was reported to be only a few thousand photons per mega electron volt (ph MeV$^{-1}$)[9] while metal halide perovskite nanocrystals achieved LYs in the range of 1000–20,000 ph MeV$^{-16,16,17,22,23}$. These LYs are significantly lower than those of conventional inorganic scintillators, such as NaI:Tl, with LYs up to 50,000 ph MeV$^{-1}$[24]. Nanocrystals are also prone to fast degradation under high-energy irradiation[25], which further hampers their potential in practical applications. To date, low scintillation efficiencies and poor stability have rendered nanocrystals impractical for real scintillator applications.

We demonstrate here that colloidal quantum shells (QSs)[26] address deficiencies of both ceramic and nanocrystal scintillators. These QSs combine solution-processability with higher light yields and faster lifetimes than commercial scintillators. We characterize the QSs under pulsed X-ray, electron, and optical excitations to establish and correlate their radio (RL), cathodo (CL), and photoluminescence (PL) properties. When exposed to hard X-rays of 11.5 keV, the light yield of the QSs is found to be as high as 70,000 ± 13,300 (mean ± standard deviation) ph MeV$^{-1}$. Importantly, the radioluminescence rise times are shorter than 100 ps, and the decay lifetime is 2.5 ns without any significant slow lifetime component, surpassing that of fast commercial scintillators by at least an order of magnitude. Moreover, QSs show improved radiation hardness with stable scintillation under continuous exposure to high-flux X-rays with an absorbed dose larger than $10^9$ Gy. In addition, we demonstrate X-ray imaging by fabricating large-area QS films, realizing a fine spatial resolution of at least 20 μm, resolving more than 20 line pairs per millimeter (LP mm$^{-1}$).

We also conduct a correlative photophysical study to provide insights into the origin of radioluminescence in the QSs. Fluence-dependent photoluminescence measurements corroborate that multiexcitons are indeed responsible for the radioluminescence and reveal that the X-ray excited state of a QS can emit with a quantum yield as high as ~30% despite averaging ~20 excitons per particle. Such a significant radiative component of the multi-exciton decay in QSs is facilitated by suppression of the Auger-Meitner recombination[27]. The suppression of Auger-Meitner processes in QSs also minimizes heat generation, resulting in improved X-ray dose stability. In addition to radioluminescence, we also showcase bright cathodoluminescence from the QSs with lifetimes as fast as 0.3 ns. The combination of efficiency, speed, and durability make this class of nanocrystals appealing for a broad range of scintillator systems.

## Results

### Synthesis of quantum shells and their optical properties

We synthesize CdS/CdSe/CdS QSs[27] in a wurtzite crystal structure (Supplementary Figs. 1 and 2, and Methods). The CdS core of quantum shells deliberately surpasses the CdS exciton Bohr radius to achieve a quantum confinement regime solely in the CdSe quantum-well layer. Accordingly, our samples include three large CdS core sizes of 4.5 nm, 6.0 nm, and 8.2 nm. The CdS cores support a CdSe quantum shell layer with a thickness of 1.9 nm, 1.4 nm, and 1.6 nm, respectively. The CdS/CdSe core-shell structures are passivated with a final CdS shell of 2-4 nm. The total particle diameter is 12-20 nm. Exemplary electron

microscopy images of the QSs are shown in Fig. 1a-d (also, Supplementary Fig. 1). Figure 1e shows the optical absorbance and photoluminescence spectra of the QSs. The QSs have a broadband absorption in the visible to UV with photoluminescence in the range of 600 – 700 nm. The average photoluminescence lifetime, obtained via multiexponential fit under weak optical excitation, ranges from 27 ns to 110 ns depending on the thickness of the core and shell regions (Fig. 1f).

### X-ray scintillation properties of quantum shells

We characterize the scintillation properties of the QSs, including the LY, radioluminescence rise and decay times, stability, and linearity using hard X-rays at the Advanced Photon Source (APS). To estimate the LY of the QS samples, we use a comparative approach. For this, we use a reference scintillator, a cerium doped yttrium aluminum garnet (Ce:YAG), with a known LY of 22,500 ph MeV$^{-1}$ at the X-ray energy of 11.5 keV (see Methods)[28]. The X-rays are incident on all samples from the surface normal in a transmission geometry, and the RL is collected via a fiber-coupled spectrometer close to the excited front surface (Supplementary Fig. 3 for comparison of RL to photoluminescence). We normalize the RL emission in all samples with the respective X-ray attenuation, hence absorption, at 11.5 keV (see "Methods"). Figure 2a shows the attenuation-normalized radioluminescence spectra (see sample thicknesses in Supplementary Table 1 and Supplementary Fig. 4, and the RL signal strength before the normalization in Supplementary Fig. 5). The attenuation lengths are 29 μm and 85 μm for CdS and YAG, respectively[29]. The QSs exhibit a larger stopping power thanks to larger effective atomic number of CdS and CdSe in the range of 40–44[30]. With 11.5 keV X-rays, the excitation mechanism is photoelectric effect. Therefore, X-ray attenuation-normalized RL intensities are directly proportional to the LYs[11]. The QS samples exhibit a significantly higher attenuation-normalized RL signal as compared to the Ce:YAG. In particular, samples with 4.5 nm and 8.2 nm core sizes achieve more than threefold greater RL compared to Ce:YAG. Therefore, the LY of these QS samples is estimated to be as high as 70,000 ± 13,300 ph MeV$^{-1}$.

To independently confirm these LYs, we use an alternative LY characterization method, i.e., pulse height spectrum. Using a $^{55}$Fe radiation source, we measure the LY of the same QS samples to be 80,000 ± 8,600 ph MeV$^{-1}$ (Supplementary Note 1 and Supplementary Fig. 6). The mean values of LYs obtained by two different methods agree within 15%. Pulse height measurements also show the single X-ray photon sensitivity of the QSs. To contrast the LY of the QSs, we also measure the LY of a conventional core/shell CdSe/CdS quantum dot sample using the same comparative approach. The core/shell quantum dot shows a much smaller LY, around 1000 ph MeV$^{-1}$, consistent with prior reports[9]. Our LY characterization method also showed a much smaller LY (6,700 ph MeV$^{-1}$) when measuring a core/gradient-shell quantum dot structure[31]. This indicates that the QSs show a substantial improvement in scintillation efficiency. The origin of this enhancement will be discussed later. Uncertainty (~20%) in LY estimates is higher than in bulk scintillators because of uncertainties in the thickness and density of the thin film QS samples. Despite that, mean values of the LYs in QSs compare well among commercial and non-commercial inorganic, organic, perovskite, and nanocrystal-based scintillators at room temperature (see Fig. 2b, Supplementary Table 2).

Another important aspect of a QS scintillator is its fast operation speed. In conventional inorganic scintillators, scintillation decay is typically on the order of 50–1000 ns, along with rise times of a few nanoseconds[32]. Millisecond or longer afterglow also occur in such materials, which is detrimental to timing and fast imaging applications. In the QSs, the RL decay is fast with 1/e decay time constants varying from 2.2 ns to 3.2 ns (Fig. 2c). Multi-exponential fitting of the RL decays showed three components of ~1 ns, ~5 ns, and ~20 ns (Supplementary Table 3). Importantly, there is no discernible slow lifetime component

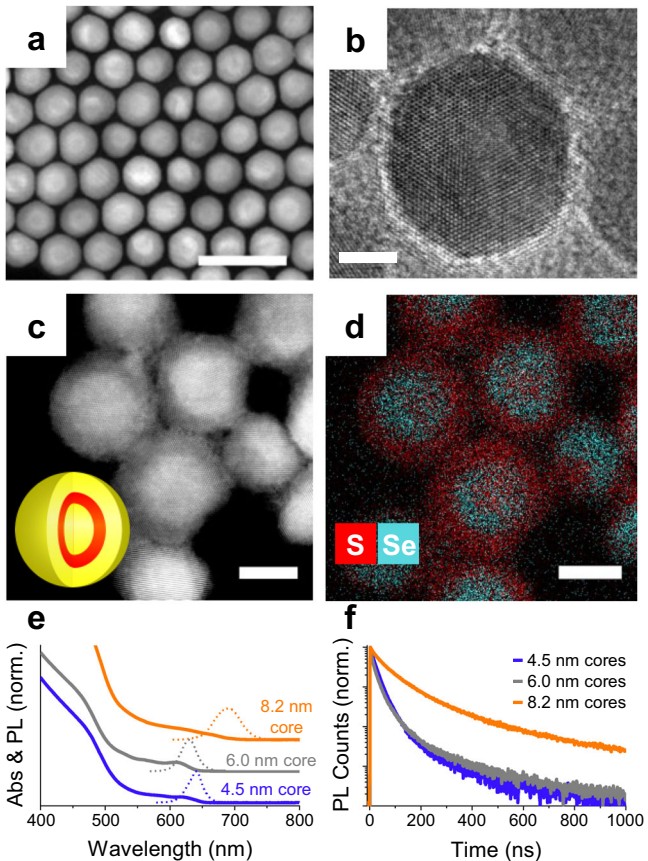

**Fig. 1 | Structural and optical properties of quantum shells.** Dark field transmission electron microscopy images (**a**, **c**) of quantum shells (QSs). The scale bar in (**a**) is 40 nm. High-resolution transmission electron microscopy image of a single QS (**b**). Scale bar in (**b**) is 5 nm. Inset of (**c**) shows a cartoon model of the QSs. The yellow region represents CdS, and the red region is CdSe. **d** Energy-dispersive X-ray mapping of sulfur and selenium from the same region as (**c**). Scale bars in (**c**, **d**) are 10 nm. Red represents sulfur, and blue represents selenium. **e** Linear absorption (solid) and photoluminescence (dotted) spectra of three QS samples with 4.5, 6.0, and 8.2 nm cores. **f** Time-resolved photoluminescence decays of three different QS samples under weak 405 nm excitation. Source data are provided as a Source Data file.

beyond 20 ns, and the RL intensity drops «1% of its initial intensity at 150 ns (Fig. 2c), consistent with ref. 18. The rise time of the radioluminescence is also fast, -100 ps (Supplementary Fig. 7), limited by the pulse-width of the X-rays at the APS.

In the past, a speed-efficiency figure of merit (M) has been defined for scintillators.

$$M = \frac{LY}{\tau} \qquad (1)$$

$M$ is the ratio between the LY and the scintillation decay lifetime ($\tau$)[33]. $M$, in units of ph MeV$^{-1}$ ns$^{-1}$, has been found to be 1,500 – 3,000 for typical ceramic scintillators[33]. Remarkably, the QSs achieve $M$ of 34,000 ± 6,500 ph MeV$^{-1}$ ns$^{-1}$ thanks to the convergence of high RL efficiency with fast operation speed. $M$ of the QSs is better than fast commercial organic crystals[34], and on par with the best reports in halide perovskites measured at cryogenic temperatures[35,36]. With its room temperature LY, the QSs set a benchmark among fast scintillators with decay lifetimes faster than 20 ns. With ultrafast response times and high radioluminescence efficiency, the QSs may be promising for timing-based applications, such as TOF-PET[37,38].

Linearity and stability are other important attributes of a scintillator. RL signal of the QSs shows a highly linear response as a function of incoming X-ray flux (Fig. 2d, Supplementary Fig. 8). The scintillation stability of the QSs is also improved as compared to other nanocrystal or emerging scintillators. Figure 2e shows the RL intensity of the QSs measured over 8 hours of continuous exposure to a substantially large flux (10$^{12}$ X-ray photons per second on 300 μm by 1 mm area). This flux corresponds to an absorbed X-ray dose larger than 10$^9$ Gy over 8 hours (see "Methods"). This dose is million times larger than typical dose limits of conventional organic and quantum dot scintillators before they significantly degrade[17]. Under exposure over 8 hours, the RL of the QSs shows a slight increase in intensity. The RL brightening, which was observed in each QS sample, could be related to the photo-brightening phenomenon reported under UV excitation[39] or positive hysteresis reported in ceramic scintillators[24]. Post-characterization of the exposed region does not show any difference either in the structure or in the optical properties, so no damage is observed (Supplementary Figs. 9 and 10). On the other hand, the RL of a core/shell CdSe/CdS quantum dot drops more than 80% of its starting intensity during the same period of exposure. Also, hybrid lead halide perovskites, such as MAPbBr$_3$ and (PEA)$_2$PbBr$_4$, shows much shorter stability with RL intensity decaying within couple of minutes under the same X-ray flux (Fig. 2e). Therefore, the QSs overcome stability problems which have been common among solution-processed scintillators.

The energy resolution is another important property, which can be inferred from the full width at half-maximum of the pulse height spectrum. After deconvolving the instrumental broadening function (Supplementary Fig. 6), we find the energy resolution of the QSs to be 75% for 5.9 keV X-rays ($^{55}$Fe source). The broadened intrinsic energy resolution is likely to be due to the very small thickness (< 5 μm) of QS scintillator films. This is consistent with energy resolution broadening observed in thinner scintillators, or when using smaller X-ray energy that results in shallower absorption depths[40,41].

We also characterize the temperature dependence of the scintillation of the QSs. Figure 2f shows that the RL signal is increased roughly twofold when the QSs are cooled from 297 to 133 K. This indicates that the LY of the QSs can be increased beyond 100,000 ph MeV$^{-1}$, approaching the theoretical limit of ~202,000 ph MeV$^{-1}$. Previous work on other material systems, such as halide perovskite single crystals and epitaxial single crystal semiconductors, have also demonstrated increased RL at lower temperatures due to suppression of nonradiative recombination channels[42].

## X-ray imaging with quantum shell films
We perform imaging experiments using a table-top X-ray tube with a copper cathode (see Fig. 3a) (see "Methods"). To showcase QS scintillators in an imaging application, large area (2.2 cm by 2.2 cm), uniform films of the QSs (Fig. 3b) are prepared by mixing them with a transparent acrylic host with a thickness of - 5 μm. We successfully image various objects placed in front of the QS scintillator screen, concealed in an envelope. These objects include metal bolts, rings, a pen, and a copper TEM grid (200 mesh) (Fig. 3c).

Next, we quantitatively characterize the spatial resolving power of the QS scintillator films. Under low magnification, the QS film scintillator resolves targets with more than 5 LP mm$^{-1}$ (see Fig. 3d) when imaging a commercial resolution test target (73-NRT, SupertechX-ray, Fig. 3e). Under high magnification, the QS scintillator can resolve the highest resolution target of 20 LP mm$^{-1}$ (Fig. 3f), with a contrast ratio $\left(\frac{I_{max}-I_{min}}{I_{max}+I_{min}}\right)$ of 0.3 (see Fig. 3g). Figure 3h shows the measured modulation transfer function (MTF). The MTF is fitted with a sigmoid, showing that the maximum spatial resolving power of the QS film is about 28 LP mm$^{-1}$ for a cut-off contrast ratio of 0.2[38]. This indicates that the QS scintillator provides an excellent spatial resolving power, better than conventional inorganic scintillators (Supplementary Fig. 11), and also

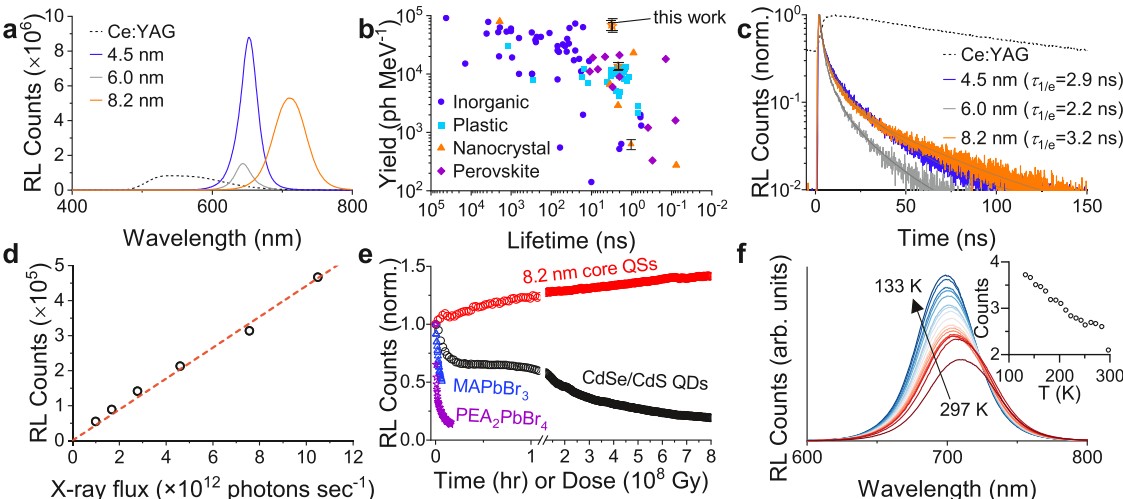

**Fig. 2 | Radioluminescence properties of quantum shells. a** Radioluminescence (RL) of three different QS samples (solid) versus Ce: YAG reference (dotted), normalized for absorbed X-ray flux in each sample. **b** Comparison of the reported lifetimes (inverted scale) and light yields of inorganic, plastic, perovskite, and nanomaterial scintillator materials. Data are gathered from literature reports or vendor websites (Supplementary Table 2). Error bars indicate standard deviation in the LY measured in our samples here. All data are for room temperature measurements. Star symbols indicate QS samples used in this work and comparison CdSe/CdS core/shell sample is shown by a triangle symbol. **c** Time-resolved radioluminescence of the QS samples and Ce:YAG reference under pulsed 11.5 keV

X-rays. **d** X-ray flux-dependence of radioluminescence of 4.5 nm core QS sample. **e** Time-dependent radioluminescence decays from QSs versus a conventional CdSe/CdS core/shell quantum dot sample and MAPbBr$_3$ and PEA$_2$PbBr$_4$ halide perovskite crystals. Measurements are taken under continuous exposure to synchrotron X-rays with an average photon flux of $10^{12}$ photons sec$^{-1}$. Conventional CdSe/CdS quantum dots and hybrid lead halide perovskites show significantly poorer stability. **f** Temperature-dependent radioluminescence from an 8.2 nm core QS sample from 297 K to 133 K. Inset shows the integrated counts versus temperature. Source data are provided as a Source Data file.

better than recent reports on thermally-activated delay fluorescence systems achieving resolution up to 20 LP mm$^{-1}$ [38,43,44]. The QS scintillators with high efficiency, rapid response time, high spatial resolution, and strong radiation hardness can be used for ultrafast dynamic imaging applications in the future.

## Radioluminescence mechanism and accompanying excited states

The absorption of an X-ray photon by a QS triggers a cascade of deexcitation processes, resulting in the generation of multiple electron-hole pairs (multiexcitons). These excitations subsequently decay through multiexciton recombination pathways, comprising both radiative mechanisms that give rise to RL, as well as non-radiative processes primarily associated with Auger-Meitner recombination, giving rise to heating[45]. To investigate the excited states responsible for the RL, we perform excitation fluence dependent photoluminescence (PL) measurements (Supplementary Fig. 12), and correlate these measurements with the time-resolved RL decays (Fig. 2c).

Figure 4a shows PL spectra from the QS sample with a 4.5 nm core in a fluence range from 0.1 to 400 μJ cm$^{-2}$, generating an average exciton population per QS, $\langle n \rangle$, ranging from 0.2 to 1000. With an increasing fluence, blue-tail of the PL peak increases due to contributions from multi-exciton recombination (Supplementary Fig. 13)[26,27,46], consistent with broadband multiexcitonic optical gain in these QSs[27]. More evidence that multiexciton emission is not merely detectable in the QSs, but also efficient, can be observed via plotting the PL intensity versus $\langle n \rangle$. Figure 4b illustrates the relative change in PL quantum yield (QY) as a function of $\langle n \rangle$. In the sample with 4.5 nm core, the QY stays roughly the same for $\langle n \rangle$ from 0.1 to 3. Then, the QY drops only to ~50% of the maximum QY, when $\langle n \rangle$ reaches 10 (see also Supplementary Fig. 14). These observations are in stark contrast with conventional quantum dots[21,47], where QY drops sharply for $\langle n \rangle$ larger than 2 since multiexciton states have diminishingly small QY. This effect is mainly due to dominant Auger-Meitner recombination in conventional

quantum dots[48–51]. Therefore, Fig. 4b illustrates that multiexciton radiative recombination is substantially efficient in the QSs.

To better understand the improved multiexciton utilization in the QSs, we measure the excitation fluence-dependence of PL decays. Figure 4c shows the buildup of a faster decay feature at higher fluences due to multiexciton recombination[52]. We extract the decay associated only with a biexciton state ($\langle n \rangle = 2$)[21] and find the corresponding biexciton Auger-Meitner lifetimes of 6.7 ns, 4.0 ns, and 11.9 ns for samples with core sizes of 4.5 nm, 6.0 nm, and 8.2 nm, respectively (Fig. 4d). These remarkably long Auger-Meitner lifetimes, in agreement with ref. 27, underpin the high QY of the multiexciton states in the QSs. Auger-Meitner lifetimes in typical quantum dot systems are on the order of 0.1 ns and could only reach ~1 ns for specially engineered gradient-shell systems[52–55]. Therefore, the QSs suppress Auger-Meitner processes, and the exciton-exciton annihilation does not dominate the multiexciton recombination.

To determine the excitonic states responsible for the RL emission, we fit the experimental RL decays by a variable-power rate scaling model, which accounts for multi-exciton population and associated emission dynamics[56]. This model allows calculating the time-dependent luminescence intensity corresponding to an average starting number of excitons per QS, $\langle n \rangle$, assuming a non-statistical scaling of radiative and Auger-Meitner rates with $n$. The input parameters for this model, as well as multiexciton rate scaling methodology, are described in Supplementary Note 2 and Supplementary Tables 4, 5. The temporal evolution of the excited state population with a fixed number of excitons per particle $\langle n \rangle$ is shown Supplementary Fig. 15. The fast decay of a large multiexciton population is followed by a slower onset of biexciton and exciton population, which explains the complex temporal decay of the RL. Figure 4e illustrates the best fit of the variable-power model to the experimental RL data for 8.2-nm-core QS, obtained using 18 excitons ($\langle n \rangle = 18$) (orange curve). For comparison, model calculations corresponding to $\langle n \rangle = 10$ (purple curve) and $\langle n \rangle = 30$ (blue curve) are also shown. This model is also consistent with an empirical approach based on a comparison of

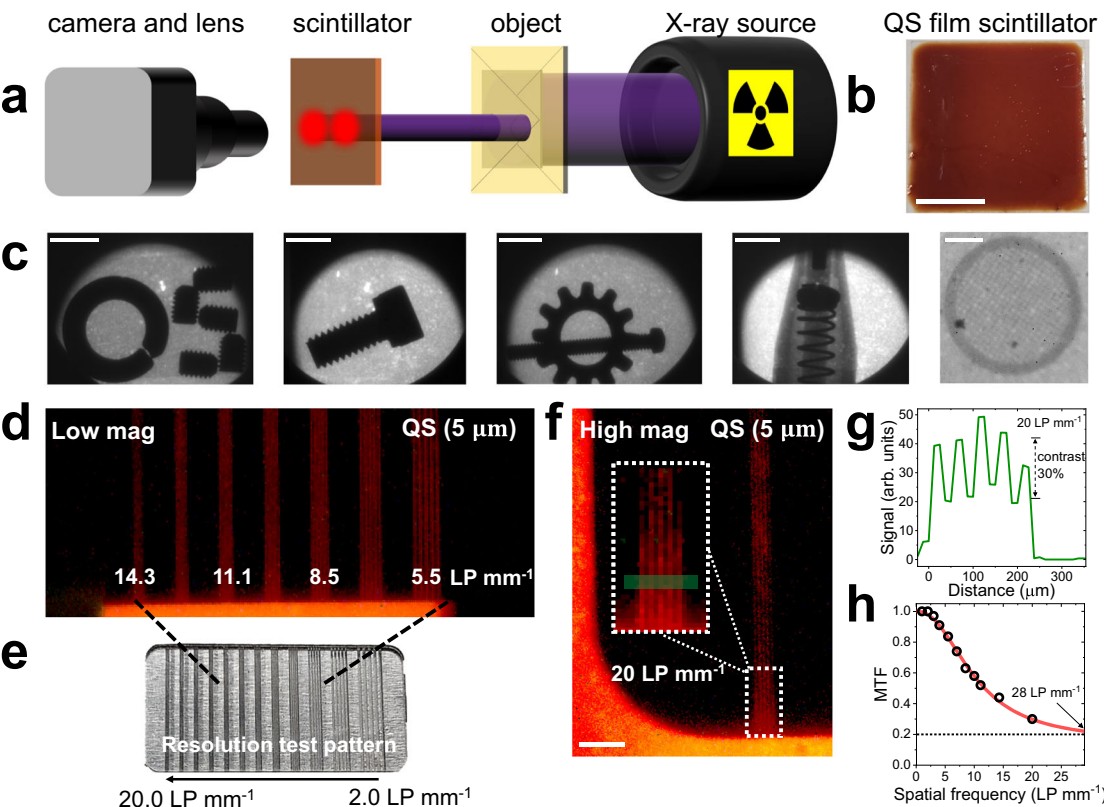

**Fig. 3 | High-resolution X-ray imaging with quantum shell thin films.**
**a** Schematic of the X-ray imaging setup using a table-top X-ray tube source and a camera with a zoom lens. Various objects are concealed in an envelope and imaged by the QS scintillator film with a thickness of 5 μm. **b** Photo of the QS - polymer scintillator film deposited on a square-shaped thin glass substrate. Scale bar is 1 cm. **c** Various objects (bolts, rings, pen and a 200-mesh copper TEM grid) were imaged by the QS scintillator with a monochromatic camera. Scale bars in (**c**) are 5 mm except the image of a TEM grid, which has a scale bare of 200 μm. **d** Low magnification imaging (with a color camera) of a resolution test chart with a QS film

scintillator. Various line pair per mm (LP mm⁻¹) values of the chart are labeled. **e** Commercial resolution test chart used here with varying resolution targets from 2.0 to 20.0 LP mm⁻¹. **f** High magnification image of 20 LP mm⁻¹ target with the QS scintillator. The scale bar is 500 μm. The inset shows a zoomed region highlighted by a white dashed rectangle. Green region in the inset of (**f**) is plotted as a line profile in (**g**). The contrast ratio for 20 LP mm⁻¹ target is 0.3. **h** Modulation transfer function (MTF) curve is measured for the QS scintillator and fitted with a sigmoidal function. At a cut-off contrast ratio of 0.2, the maximum resolving power is found to be 28 LP mm⁻¹. Source data are provided as a Source Data file.

fluence dependent $\tau_{PL}$ to $\tau_{RL}$ (Supplementary Fig. 16). This empirical approach suggests that the RL state comprises $\langle n \rangle$ in the range of 10 - 20 excitons. Therefore, both approaches consistently estimate a large number of excitons per QS, in the range of tens of excitons, contributing to the RL emission.

Having identified $\langle n \rangle$ responsible for the RL, we now quantify the QY of the corresponding multiexciton state given the measured LY. The most universal definition of a scintillator efficiency ($q$), under conservation of energy, is defined[57] by the light yield of photons emitted ($N_{ph}$) and their average energy ($\langle h\nu \rangle$), divided by the energy of incident radiation ($E_i$):

$$q = N_{ph} \langle h\nu \rangle / E_i \qquad (2)$$

However, more commonly, the following empirical relationship[33] is used to define the LY

$$LY = \frac{1000}{\beta E_g} \gamma \, QY \left( \frac{\text{photons}}{\text{keV}} \right) \qquad (3)$$

where $E_g$ is the bandgap, $\beta$ is a unitless parameter related to impact ionization threshold, $\gamma$ is a factor considering the fraction of X-ray photon energy converted into electronic energy, QY is the quantum yield of the RL state. $\beta$ is 2.6 for CdS[58], and $E_g$ is -1.9 eV for the QSs. We assume $\gamma \approx 1$ consistent with previous work in other

scintillators[38]. Under these assumptions, the maximum LY of QSs is 202,000 ph MeV⁻¹, assuming the QY is 1.

Considering the measured LY (70,000 ph MeV⁻¹ for the core size of 8.2 nm), the QY of the RL emitting state has to be 0.35. Now, we check if this QY is reasonable given the $\langle n \rangle$ estimated by the variable-power rate scaling model. This model shows $\langle n \rangle = 18$ for the RL emission (see Fig. 4e). Figure 4b shows that the QY of $\langle n \rangle = 18$ is 50% of the QY of $\langle n \rangle = 1$ in the sample with core size of 8.2 nm. Given that the QY of a single exciton state ($\langle n \rangle = 1$) is 0.4 - 0.6[27], the QY of $\langle n \rangle = 18$ would be 0.2 - 0.3. Therefore, the QY of the RL emitting state is self-consistent and this further validates the high LYs measured here. In conventional quantum dots, the QY of $\langle n \rangle > 2$ is diminishingly small (« 0.1) as multiexcitons recombine via Auger-Meitner process[48]. This then leads to poor LY in conventional quantum dots. Therefore, suppression of Auger-Meitner recombination in QSs is the key to their achievement of large LYs.

Number of scintillation photons emitted by a single QS (= QY × $\langle n \rangle$ ≈ 5.3 ph) is lower than the total collected RL photons (= LY × $E_{X-ray}$ ≈ 805 ph) per absorbed X-ray photon (Supplementary Note 3). This implies that an absorbed X-ray photon causes a considerable energy deposition across multiple QSs. Based on the estimated $\langle n \rangle$, QY of the RL state and the measured LY, we calculate the spatial spread to be across ~150 QSs at 11.5 keV X-ray energy in a close-packed QS film. This spread corresponds to a volume with a diameter of ~126 nm (Supplementary Note 3, Supplementary Fig. 17). This spread

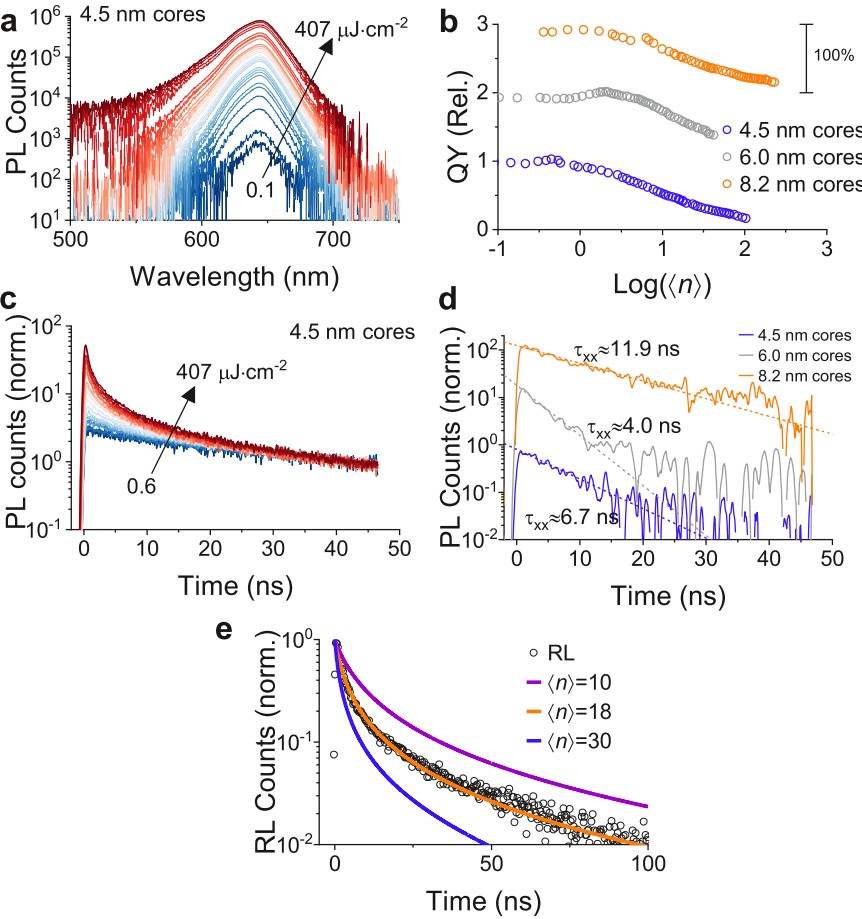

**Fig. 4 | Uncovering excited electronic states responsible for scintillation.**
**a** Excitation fluence-dependent photoluminescence spectra of QS sample with
4.5 nm core. Arrow indicates the growth of intensity with increasing fluence.
**b** Comparison of integrated photon counts versus the logarithm of the average
number of electron-hole pairs generated per QS, $\langle n \rangle$. Data is shown for both 4.5, 6.0
and 8.2 nm core QS samples. **c** Early-time decay of photoluminescence of the
4.5 nm core QS sample as a function of fluence. **d** Differenced fluence-dependent
time-resolved photoluminescence data to extract biexciton lifetime for all three QS
samples. Differenced data are generated by normalizing high- and low-fluence data
at a long delay time (as in **c**) and subtracting the low-fluence data from that of
higher-fluence. In order to extract biexciton lifetimes, an average number of
electron-hole pairs of the higher-fluence case is kept at less than 1 to ensure minimal
contribution from higher-order excitonic species. Dashed lines indicate an expo-
nential fit. **e** Power-scaling model calculation of the emission intensity decay
resulting from multiexciton populations in a 8.2 nm core QS sample. The orange
curve represents the best model fit of the experimental RL decay data using $\langle n \rangle = 18$.
Model calculations corresponding to $\langle n \rangle = 10$ exciton (purple) and $\langle n \rangle = 30$ exciton
populations (blue) are also shown. Source data are provided as a Source Data file.

is consistent with secondary electron scattering and deposition,
extending up to tens to hundreds of nanometers[59]. Additionally, near-
field (e.g., Förster) energy transfer processes may contribute to this
spread in the QS films[60].

Finally, we show that the QSs exhibit bright, ultrafast cath-
odoluminescence under energetic electron irradiation (Supplemen-
tary Note 4). The CL lifetimes ( ~ 0.3 ns) are even faster than that of RL,
while the CL signal is considerably stronger than a gold CL calibration
sample (Supplementary Figs. 18–19). Further work is under progress to
understand the faster CL decays, and the potential effects of sample
charging under electron irradiation.

In the quest for scintillators that can simultaneously embody
efficiency, speed, durability, and other desirable properties, we intro-
duce colloidal quantum shells as a promising contender. These
materials exhibit scintillation light yields that surpass the best inor-
ganic scintillators while offering unprecedented high-speed operation
with decay times as short as 2 ns with sub−100 ps rise times. Notably,
quantum shells demonstrate significant resilience even under
unusually intense X-ray irradiation, surpassing the longevity of other
perovskite- and quantum dot-derived scintillators by six orders
of magnitude larger X-ray dose resilience. QS scintillators also offer
high stopping powers and maintain consistent linearity in their

performance. Moreover, QS scintillator films empower high-resolution
imaging (28 LP mm$^{-1}$). By harnessing the potential of these quantum
shells in practical scintillators, one can envision applications such as
super-resolution dynamic multiplexed imaging and X-ray-triggered
biochemistry. As we continue to engineer and explore the capabilities
of quantum shells, their large-area film formation and compatibility
with flexible substrates offer exciting avenues for future advancements
in broader scintillator technologies.

## Methods
### Materials for the synthesis
The chemicals described below are used without further purification.
We also did not apply any modification to chemicals. These chemicals
include anhydrous acetone (99 %, Amresco), cadmium oxide (CdO,
99.95%, MilliporeSigma), Anhydrous ethanol (99%, BeanTown Chemi-
cal), hexane (ACS grade, Thermo Scientific), 1-octadecene (ODE,
technical grade, 90%, MilliporeSigma), Octane (98%, MilliporeSigma),
1-octanethiol (97%, Alfa Aesar), oleic acid (OA, technical grade, 90%,
MilliporeSigma), oleylamine (OLAM, technical grade, 70%, Milipor-
eSigma), poly(butyl-co-isobutyl)methacrylate (Sigma, $M_W$ ~ 354000),
Dioctylamine(DOA, 97%, MilliporeSigma), selenium powder (Se, 99.5%,
200 mesh, Thermo Scientific), sulfur powder (S, 99.999%, Thermo

Scientific), toluene (99.8%, MiliporeSigma), and tri-n-octylphosphine (TOP, 97%, Strem Chemical).

## Synthesis of 6−9 nm CdS quantum dot cores

Bulk-sized (6−9 nm) CdS cores were synthesized by following a coalescence-based growth[61]. We loaded 8 mL OLAM and 42 mg $CdCl_2$ into a 25 mL flask. The solution was kept under an argon atmosphere in a Schlenk line. The solution was then heated to 240 °C. After stabilization of the temperature, 540 nmols of small CdS seeds (2-4 nm diameter) were quickly injected into the flask. The solution was left for coalescence and growth for 60 minutes. After this time, the solution was quickly quenched by a water bath. The CdS cores were cleaned two times by precipitating the nanocrystals in a toluene/ethanol mixture. The cleaned CdS cores were dissolved in a mixture of 5 mL of ODE and 8 mL of OA. The mixture was then loaded into a flask, which is in an argon atmosphere. The flask was heated to 150 °C for 60 minutes. After that, the flask was cooled to room temperature. The CdS cores were then precipitated twice as in the previous step. In the final step, CdS cores were dissolved in hexane and stored at room temperature. This recipe produces CdS quantum dots with a diameter ranging from 6 nm to 9 nm. CdS quantum dot cores with a larger size (10 nm -12 nm in diameter) were also synthesized by raising the temperature to 270 °C, and using a doubled small CdS seed concentration (1080 nmols).

## Synthesis of CdS/CdSe core/shell quantum dots

A thin shell of CdSe was grown onto the CdS cores by injecting two 0.1 M Cd-oleate and 0.1 M TOP-Se precursors using a syringe pump. To note, these precursors were not mixed before the injection. The Cd-oleate precursor includes 412 mg CdO, 8 mL OA, and 5 mL ODE mixed in a 50 mL flask. The flask was then heated to 260 °C under an argon environment. The precursor was ready once the solution became clear and nearly colorless. Once this condition is reached, 19 mL of ODE was injected into the same flask. The Se precursor includes 141 mg Se powder and 3 mL TOP mixed in a 25 mL flask. The Se precursor-containing flask was then heated to 140 °C under an argon environment. We waited for all the selenium powder to react. After that, 14 mL of ODE was injected into the same flask for the purpose of dilution. CdSe shell growth started by loading CdS cores (4 nm - 9 nm in diameter) into a 100 mL flask. To this flask, 2 mL of dioctylamine (DOA) and 2 mL ODE were also added. The solution in the flask was degassed to remove the remaining water and the temperature was slowly increased to 110 °C while the bubbling was finished. After this step, the mixture in this flask was transferred to a Schlenk line, and the gas flow environment was switched to argon. We set the temperature to 315 °C. While the temperature was rising, we began injecting the Cadmium and Selenium precursors at 270 °C. The rate of injection was chosen as 3 mL hr$^{-1}$. We continued the injection process as the peak emission wavelength of the core/shell quantum dots reached the wavelength range, typically from 630 nm to 680 nm. Total injection time was typically between 90 min to 100 min for obtaining larger core/shell quantum dots. For medium size core/shell quantum dots, injection time was shorter (~ 70 min). As the injection was finished, we removed the flask from the heating mantle and let the solution cool down to room temperature. The as-synthesized core/shell quantum dots were cleaned by precipitating the solution within a mixture of toluene and ethanol/acetone (1/2). The cleaned core/shell quantum dots were dissolved in hexane.

## Synthesis of CdS/CdSe/CdS quantum shells

Core/shell CdS/CdSe quantum dots were loaded into a flask (volume of 100 mL). In the same flask, 2 mL DOA and 2 mL ODE were also added. This flask was heated to 120 °C and degassed to remove water and other contaminants. The flask was then connected to a Schlenk line, and the gas environment was changed to argon. The temperature of the flask was increased to 315 °C. We prepared 0.1M Cd-oleate and loaded it into a syringe pump. Also, we prepared 0.12 M ocatnethiol-ODE prepared by mixing 0.34 mL of octanethiol with 11.66 mL of ODE. This solution was loaded into a different syringe pump. Both syringe pumps were injected into the flask containing core/shell quantum dots with an injection rate of 3 mL hr$^{-1}$. The injection was started when the flask reached a temperature of 270 °C. We stopped the core/shell/shell quantum dot, i.e., quantum shell, growth once the shell thickness reached a desirable level. The growth took 360 minutes for the 6.0 nm core sample and 220 minutes for 8.2 nm core sample. When we finished the injection, we left the quantum shell solution in the flask at 315 °C for 45 minutes. This was done intentionally for annealing the samples. After that, we cooled the solution to room temperature. We cleaned the quantum shell samples by precipitating them in toluene and a ethanol:acetone (1:2) mixture under vigorous centrifugation. We dissolved the quantum shells in hexane.

## Optical Spectroscopy

Absorption spectra of the samples were collected using a Lambda UV-Vis-NIR spectrometer. Photoluminescence data was collected using a pump generated from a frequency-doubled 2 kHz Ti: sapphire laser (800 nm doubled to 400 nm), with emission collected by fiber and directed onto a CCD array. Time-resolved emission was collected using the same configuration but directed instead to a streak camera (Hamamatsu). The index of refraction for the films was estimated from thin films spin-coated onto silicon using an Alpha-SE ellipsometer using a Cauchy model of the transparent region. This model was then applied to determine the film thickness of samples using a Filmetrics optical profilometer.

## Radioluminescence characterizations

Static and time-resolved radioluminescence properties of the samples are characterized using the time-resolved X-ray induced optical luminescence (TR-XEOL) spectroscopy setup at the Beamline 25-ID-E at the Advanced Photon Source (APS). Briefly, the pulsed X-rays out of the APS in 24-bunch mode were used as the excited source. The X-ray pulse repetition rate is 6.536 MHz. Radioluminescence was collected by a large core multimode optical fiber (1.5 mm core diameter) with a large numeric aperture (NA = 0.54) achromatic fiber collimator. The fiber was input into an Andor Shamrock 303i spectrograph with a grating of 150 ln mm$^{-1}$. The static spectra were captured by iDUS 420 series Andor camera at one of the two exit ports of the spectrograph. The other exit port was used for the time-resolved radioluminescence measurements using a single photon avalanche detector (Micro Photon Devices, MPD) with a 50 ps intrinsic time resolution. A time-correlated single photon counting unit (Picoquant, PicoHarp300) converts the SPAD output into radioluminescence decays while being synchronized to the bunch repetition rate of 6.536 MHz, or the ring repetition rate (6.536/24 = 272.3 kHz). The decay curves are fitted with exponential functions using Origin Pro software. The X-ray flux is adjusted by a set of aluminum filters, and the X-ray flux is measured by an ion chamber and a large-area calibrated PIN diode.

## Comparative light yield calculation

Light yields of QS films are calculated by comparison to a standard scintillator of Ce: YAG, a commercial sample by Crytur, with a light yield of 22,500 ph MeV$^{-1}$ at 11.5 keV, as confirmed by the vendor. The thickness of the standard is 100 μm; sample thicknesses are measured using optical profilometry and range from 1–5 μm. Light yield (LY) estimates are made based on measurements performed at normal incidence with no change in the collection geometry, time, monochromator, slits, or camera. The CCD camera has ~90 % efficiency

throughout the studied spectral range. The comparative light yield measurement is made in the most generic form using the formalism of comparative quantum yields:

$$LY_x = LY_r \left( \frac{1 - e^{-t_r/\alpha_r}}{1 - e^{-t_x/\alpha_x}} \right) \left( \frac{D_x}{D_r} \right) \left( \frac{n_x}{n_r} \right)^2 \qquad (4)$$

In which subscript r denotes the reference (Ce:YAG), $\alpha$ is the attenuation length necessary for X-ray intensity to be reduced to $1/e$, $t$ is the thickness, $D$ is the integrated area under the emission curve, and $n$ is the index of refraction. Attenuation lengths are calculated using a CXRO[29] database based on elemental composition and density. When calculating the attenuation length of the QS composites, we assume that CdS (which dominates the composition) represents only 65% of the total volume of the QS solid, based upon close-packing of random spheres in addition to organic ligands. Additionally, in practice, the index of refraction of Ce: YAG is almost identical to that of the QS films. The uncertainty in the light yield is calculated by the error propagation method. Error in thickness is ~13% for the highest efficiency sample (4.5 nm core). Error in particle density is ~10%. The propagated error is ~19%. Propagated errors are 16% and 17% for 6.0 and 8.2 nm cores, respectively.

### Absorbed X-ray dose estimation

To estimate the absorbed dose ($D$) during the stability test (Fig. 2e), we use the following relationship. $D = F_{X-ray} \times t^{-1} \times Att \times (d_{sample})^{-1} \times time$, where $F_{X-ray}$ is the X-ray flux per sec in units of $Js^{-1}m^{-2}$, $t$ is the sample thickness, Att is X-ray attenuation amount by the sample, $d_{sample}$ is the density of the sample, time is the exposure time. At 11.5 keV energy, $10^{12}$ X-ray photons impinging on an area of 300 μm and 1 mm corresponds to a flux of $F_{X-ray} = 1.8425 \times 10^{-15} J \times 10^{12} s^{-1} \times \frac{1}{3 \times 10^{-4} \times 1 \times 10^{-3} m^2} = 6141.7 \, Js^{-1} \, m^{-2}$. $t = 5 \times 10^{-6}$ m. Att = 0.16 is based on X-ray transmission from a 5 μm thick CdS layer for 11.5 keV X-rays. $d_{sample} = 4.82 \times 10^3 \, kgm^{-3}$ for CdS.

For 1 s exposure, the absorbed dose $D = 40774.8 \, Jkg^{-1} = 40.775$ kGy. For 8 hours exposure, the absorbed dose is $1.17 \times 10^9$ Gy. As compared to conventional lab-based sources ($10^3 Gy \, hr^{-1}$)[17], the dose rate ($\sim 10^8 Gy \, hr^{-1}$) here is substantially higher.

### Radioluminescence Imaging

Imaging experiments are performed using a table-top X-ray tube source using a copper cathode. The acceleration voltage is 40 kV. Monochrome (FLIR Grashoper3) and color (FLIR BlackflyS) cameras are used with a focusing zoom lens (working distance of ~30 cm) to image the scintillator film. These quantum shell films were prepared either by drop-casting onto a mercaptopropyltrimethoxysilate-treated glass, or by dispersing the QSs into a 5 wt% mixture of poly(butyl-co-isobutyl) methacrylate in chloroform, then casting the dispersion on to a thin glass slide for it to dry. X-ray beam excites the quantum shell-coated side of the glass piece. The camera and objective images of the scintillation through the transparent glass side.

### Physical Characterization

Powder x-ray diffraction is collected on the QS samples by drop-casting thick films on to mis-cut silicon substrates and measured using a Bruker D2 phaser tool. Electron microscopy is performed on dilute drop-cast films using a JEOL 2100 F and ThermoFisher Scientific Spectra 200. Dense films are imaged using a JEOL IT800HL SEM.

### Cathodoluminescence measurements

Cathodoluminescence measurements are performed in an FEI Quattro SEM using a 0.97 NA parabolic mirror and a Delmic Sparc cathodoluminescence module. Spectra are acquired on an Andor Kymera spectrograph with an Andor Newton CCD. Time-resolved CL is acquired by pulsing the FEG using the third harmonic of a Mai:Tai Ti:sapphire fs laser oscillator and collecting CL counts on Quantum Opus large area superconducting nanowire single photon detectors with detected photon events time-tagged using a PicoQuant Hydraharp.

## Data availability

Other datasets generated and/or analyzed during the current study are available from the corresponding author on request. Source data are provided in this paper.

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

## Acknowledgements

We would like to thank Dmitri Talapin, Igor Coropceanu, Josh Portner, Aritrajit Gupta for providing colloidal quantum dot samples for initial radioluminescence tests at the Advanced Photon Source. We also thank Antonino Micelli, Donald Walko, and Xiaoyi Zhang for fruitful discussions on radioluminescence imaging and pulse height spectrum character-izations. Work performed at the Center for Nanoscale Materials and Advanced Photon Source, both U.S. Department of Energy Office of Science User Facilities, was supported by the U.S. DOE, Office of Basic Energy Sciences, under Contract No. DE-AC02-06CH11357. The work of the BGSU team was supported by the Award DE-SC0016872 funded by the U.S. Department of Energy, Office of Science. Cathodolumines-cence measurements were supported by the Center for Nanophase Materials Sciences (CNMS), which is a US Department of Energy, Office of Science User Facility at Oak Ridge National Laboratory.

## Author contributions

B.G. performed steady-state and time-resolved X-ray-induced radi-oluminescence and pulse height characterizations. B.G. and B.T.D. performed data analysis on radioluminescence data. B.T.D. and M.H. performed transient optical measurements and analyzed the data. B.G. performed X-ray imaging experiments and the associated data analysis. J.P.C., D.H., and M. Z. synthesized the quantum shells. M. Z. supervised the synthesis effort. B.T.D. produced thin film samples and performed film thickness measurements and microscopy imaging. M. Z. developed and applied a variable-power exciton model. R.D.S. provided perovskite comparison samples. X.M.L. performed inductively coupled plasma measurements for estimating particle density. B.L. and V.I. performed cathodoluminescence measurements. B.G., B.T.D., and M.Z. wrote the paper with contributions from all authors.

## Competing interests

B.G., B.T.D., and M.Z. have a joint patent application under review that describes the use of quantum shells for scintillators. All other authors declare no competing interests.
