## [Peer Review File · Nature Communications]

REVIEWER COMMENTS

Reviewer #1 (Remarks to the Author):

I find that the manuscript has been properly revised to account for my and other reviewers' comments. I recommend it for publication

Reviewer #2 (Remarks to the Author):

I would like to thank the authors for the work done and for the clarifications received. The results obtained are excellent and the work is certainly very interesting.

However, I still have some doubts related to the emphasis given to light yield and the comparison made with standard scintillators.

The evaluation of light yield in this type of scintillator is certainly very delicate and complex. The authors have done a remarkable work using two different methods for estimating the light yield. However, the uncertainties on the evaluation of the light yield are very large. For example, as reported by the authors themselves, there is a difference of 15% between the evaluation of light yield with RL comparative measurements and that with the pulse height spectrum. Moreover, for what concern the RL measurement alone, there is an uncertainty of >10% on the thickness of the QS film (i.e. sample core: 4.5 nm, thickness 3429+/-444) which, through normalization, propagates on the estimate of the light yield. Moreover, also the uncertainty on the QS film density must be taken into consideration.

This is just an example to underline how difficult is the estimate of the light yield for these samples and, consequently, it is very tricky and risky to make a direct comparison with other scintillators as is done several times in the article and reported for example in Fig. 2b and Table S3. If the authors believe it is essential to make this type of comparison, I think it's better to report also the uncertainty and not only the light yield mean values or at least insert some comments on the accuracy of the measurements.

Linked to this is the discussion on the energy resolution of 75% obtained in the measurement with the ⁵⁵Fe source reported on page. 8. For a scintillator with such a high light yield I would expect a significantly better energy resolution. Maybe an explanation for the low energy resolution obtained could be related to the fact that having a very thin film (only 5 μm) the electrons produced by the photoelectric effect can release an ever-changing fraction of their energy in the scintillator thus producing a broadening of the energy resolution. This is another example of how delicate it is to make a comparison between the energy resolution (and the light yield) of a standard 'bulk' scintillator and such a thin film.

Finally, there are some aspects that need to be double-checked within the text:

- Several references are wrong such as on page. 6 "sample thicknesses in Table S1" -> Table S2 or "see SI Section 1 and Figure S6" -> S8
- Some citations are reported several times in References
- In Methods – Comparative light yield calculation, if alpha is the attenuation length I think that in the formula for the comparative quantum yields the exponent should be t/α

Our responses are provided below together with all modifications made in the revised manuscript. Reviewer comments are marked with yellow highlights, while changes implemented in the revised manuscript are indicated with green. Text without highlights represents our responses.

Reviewer #1 (Remarks to the Author):

I find that the manuscript has been properly revised to account for my and other reviewers' comments. I recommend it for publication

We thank the Reviewer for positive evaluation and recommending our work for a publication.

Reviewer #2 (Remarks to the Author):

I would like to thank the authors for the work done and for the clarifications received. The results obtained are excellent and the work is certainly very interesting.

We thank the Reviewer for supportive remarks on our work.

However, I still have some doubts related to the emphasis given to light yield and the comparison made with standard scintillators.

We appreciate the Reviewer for pointing out the challenge of precise determination of light yield in ultrathin film scintillators. Following the Reviewer's suggestions, we amended this aspect in the revised manuscript by reporting uncertainties associated with LY estimates, and reducing the emphasis on the LY comparison with bulk scintillators (see our response below).

The evaluation of light yield in this type of scintillator is certainly very delicate and complex. The authors have done a remarkable work using two different methods for estimating the light yield. However, the uncertainties on the evaluation of the light yield are very large. For example, as reported by the authors themselves, there is a difference of 15% between the evaluation of light yield with RL comparative measurements and that with the pulse height spectrum. Moreover, for what concern the RL measurement alone, there is an uncertainty of >10% on the thickness of the QS film (i.e. sample core: 4.5 nm, thickness 3429+/-444) which, through normalization, propagates on the estimate of the light yield. Moreover, also the uncertainty on the QS film density must to be taken into consideration. This is just an example to underline how difficult is the estimate of the light yield for these samples and, consequently, it is very tricky and risky to make a direct comparison with other scintillators as is done several times in the article and reported for example in Fig. 2b and Table S3. If the authors believe it is essential to make this type of comparison, I think it's better to report also the uncertainty and not only the light yield mean values or at least insert some comments on the accuracy of the measurements.

We express our gratitude to the Reviewer for recognizing our effort in estimating the light yield of quantum shell thin film scintillators. We agree that accurately gauging the light yield of an ultrathin film scintillator presents a challenge, so it is important to determine experimental uncertainties. Following the Reviewer's suggestions, we now report uncertainty in light yield measurements and incorporate

these in the main text and SI. Furthermore, in accordance with the recommendation, we have reduced the emphasis in comparing the LYs of Qs with those of bulk scintillators throughout the revised text.

In the abstract: “Quantum shells exhibit superior room-temperature X-ray scintillation, with light yields up to 70,000 photons per MeV ~~surpassing the best commercial inorganic scintillators.~~”

On page 3: “When exposed to hard X-rays of 11.5 keV, the light yield of the Qs is found to be as high as $70,000 \pm 13,300$ (mean \pm standard deviation) ph/MeV, ~~better than most commercial ceramic scintillators.~~”

On page 6: “Therefore, the LY of these QS samples is estimated to be as high as $70,000 \pm 13,300$ ph/MeV.”

“~~To independently confirm~~ these large LYs, ~~we use~~ an independent method, i.e., pulse height spectrum. Using a ^{55}Fe radiation source, we measure the LY of the same QS samples to be $80,000 \pm 8,600$ ph/MeV (see SI Section 1 and Figure S6). The ~~mean values of~~ LYs determined with two independent methods agree within 15%.”

“~~Our LY characterization method also showed much smaller LY (6,700 ph/MeV) when measuring a core/gradient-shell quantum dot structure.~~³¹”

“~~Uncertainty (~20%) in LY estimates are higher than bulk scintillators because of uncertainties in thickness and density of the thin film QS samples. Despite that, mean values of the LY in Qs compare well among commercial and non-commercial inorganic, organic, perovskite and nanoparticle based bulk scintillators at room temperature (see Figure 2b, Table S3). Figure 2b compares the LY of the Qs (marked with stars) to various commercial and non-commercial inorganic, organic, perovskite and nanoparticle-based scintillators reported at room temperature (see Table S2 for the source data). The LY of Qs is on par with the best reported LY levels of bulk scintillators at room temperature.~~”

On page 7: “Qs achieve M of $34,000 \pm 6,500$ ph/MeV/ns”

On page 16, Methods section: “~~The uncertainty in the light yield has been calculated by error propagation method. Error in thickness is ~13% for the highest efficiency sample (4.5 nm core). Error in particle density is ~10%. The propagated error is ~19%. Propagated errors are 16% and 17% for 6.0 and 8.2 nm cores, respectively.~~”

We also updated Figure 2b with error bars representing the standard deviation in each QS sample.

~~Linked to this is the discussion on the energy resolution of 75% obtained in the measurement with the ^{55}Fe source reported on page. 8. For a scintillator with such a high light yield I would expect a significantly better energy resolution. Maybe an explanation for the low energy resolution obtained could be related to the fact that having a very thin film (only 5 μm) the electrons produced by the photoelectric effect can release an ever-changing fraction of their energy in the scintillator thus~~

producing a broadening of the energy resolution. This is another example of how delicate it is to make a comparison between the energy resolution (and the light yield) of a standard 'bulk' scintillator and such a thin film.

We thank the Reviewer for this comment indicating another challenge in very thin scintillator samples. As the Reviewer indicates, the broad energy resolution of QS ultrathin scintillator is likely to be due to small film thickness. Previously, the effect of film thickness has been considered for energy resolution of ceramic scintillators. Small thickness and usage of lower X-ray energy (*i.e.*, shorter penetration depth) were both shown to broaden intrinsic energy resolution (Ex./ [https://doi.org/10.1016/S0168-9002\(01\)01912-X](https://doi.org/10.1016/S0168-9002(01)01912-X) and <https://doi.org/10.1109/TNS.2007.908580>). Thus, sub-5 μm film thickness of QS scintillator films may negatively impact the energy resolution of the quantum shells. We acknowledge this potential shortcoming in the revised main text and cite previous literature.

On page 8: "Broadened intrinsic energy resolution is likely to be due to very small thickness (< 5 μm) of the QS scintillator films. This is consistent with energy resolution broadening observed in thinner scintillator films, and when using smaller X-ray energies resulting in shallower absorption depths.^{40,41}"

Finally, there are some aspects that need to be double-checked within the text:

- Several references are wrong such as on page. 6 "sample thicknesses in Table S1" -> Table S2 or "see SI Section 1 and Figure S6" -> S8

Supplementary Figure & Table numbers are cross-checked and referrals are updated.

- Some citations are reported several times in References

Citation list is checked and repeating citations are corrected.

- In Methods – Comparative light yield calculation, if alpha is the attenuation length I think that in the formula for the comparative quantum yields the exponent should be t/α

We updated the equation.

We are grateful for the feedback from all Reviewers, which have helped us to further improve our work. We thank all Reviewers for careful reading, and we hope that these revisions will fully resolve their requests.

REVIEWERS' COMMENTS

Reviewer #2 (Remarks to the Author):

I would like to thank the authors for the work done. I find that the manuscript has been properly revised and I recommend it for publication.